# An Oil-Based Adjuvant Improves Immune Responses Induced by Canine Adenovirus-Vectored Vaccine in Mice

**DOI:** 10.3390/v15081664

**Published:** 2023-07-30

**Authors:** Manon Broutin, Fleur Costa, Sandy Peltier, Jennifer Maye, Nicolas Versillé, Bernard Klonjkowski

**Affiliations:** 1UMR Virologie, INRAE, ANSES, EnvA, 94700 Maisons-Alfort, France; manon.broutin@inserm.fr (M.B.); fleur.watier@vet-alfort.fr (F.C.); 2SEPPIC Paris La Défense, 92250 La Garenne Colombes, France; sandy.peltier@inserm.fr (S.P.); jennifer.maye@airliquide.com (J.M.); nicolas.versille@airliquide.com (N.V.)

**Keywords:** canine adenovirus, recombinant vaccine, oil adjuvant, RT-PCR array

## Abstract

There is a significant need for highly effective vaccines against emerging and common veterinary infectious diseases. Canine adenovirus type 2 (CAV2) vectors allow rapid development of multiple vaccines and have demonstrated their potential in animal models. In this study, we compared the immunogenicity of a non-replicating CAV2 vector encoding the rabies virus glycoprotein with and without Montanide^TM^ ISA 201 VG, an oil-based adjuvant. All vaccinated mice rapidly achieved rabies seroconversion, which was associated with complete vaccine protection. The adjuvant increased rabies antibody titers without any significant effect on the anti-CAV2 serological responses. An RT^2^ Profiler™ PCR array was conducted to identify host antiviral genes modulated in the blood samples 24 h after vaccination. Functional analysis of differentially expressed genes revealed the up-regulation of the RIG-I, TLRs, NLRs, and IFNs signaling pathways. These results demonstrate that a water-in-oil-in-water adjuvant can shape the immune responses to an antigen encoded by an adenovirus, thereby enhancing the protection conferred by live recombinant vaccines. The characterization of early vaccine responses provides a better understanding of the mechanisms underlying the efficacy of CAV2-vectored vaccines.

## 1. Introduction

Recombinant live vaccines have emerged as a highly valuable vaccine platform, owing to their capability to transfer antigen-encoding genes into host cells and stimulate targeted immune responses in vaccinated individuals or animals. This convenient technology enables host cells to express a wide range of antigens and facilitates the design of multi-serotype or marker vaccines. Moreover, it enables the rapid development and production of multivalent vaccines from a single vectored system, providing an undeniable advantage in response to emerging infectious disease outbreaks.

Adenovirus-based vectors are a highly attractive option for vaccination purposes due to their ability to promote robust intracellular antigen expression, leading to CD8^+^ T cell-biased responses. In addition, they have been shown to induce antigen-specific CD4^+^ T cell responses and antibodies in mammalian hosts [1] and have been found to provide better protection than poxvirus or plasmid DNA vectors [2]. Adenovirus-vectored vaccines against infectious diseases have demonstrated advantageous immunogenicity and tolerability for both human and veterinary applications, as evidenced by their recent approval for use in humans to prevent diseases caused by Ebola virus (Zabdeno) or SARS-CoV-2 (COVID-19 Vaccine Janssen and Vaxzevria).

Recombinant vaccines for veterinary applications have been developed using canine adenovirus type 2 (CAV2), which has demonstrated its efficacy in mammalian hosts. Proof of concept has been shown in sheep [3], pigs [4], and goats [5] using CAV2-based vectors designed to prevent a wide range of virus-induced diseases, including rabies [6,7], porcine reproductive and respiratory syndrome [4], bluetongue [3], and foot and mouth disease [8].

Adjuvants are critical components of vaccines, helping to enhance immune responses and improve long-term protection. Promising results have been reported from vaccination trials that combine human adenovirus type 5 (Ad5) with Montanide^TM^ ISA 206 VG (ISA 206), a stable, safe, and fluid water-in-oil-in-water emulsion that enhances short- and long-term immune responses against various antigens [9,10,11]. However, our current understanding of the initial molecular interactions between the vaccine and the host, as well as the subsequent innate immune responses that contribute to the advantages of the adjuvanted adenoviral vectors, remains limited.

In order to enhance cellular responses, Montanide^TM^ ISA 201 VG (ISA 201) was developed as an improved version of ISA 206. Both adjuvants are safe, efficient, and readily available. Multiple vaccination protocols involving pigs [12] and cattle [13] have demonstrated that ISA 201 is superior to ISA 206. Therefore, to optimize the development of new adenovirus-based vaccines, it is crucial to evaluate the efficacy of the ISA 201 adjuvant and explore how innovative vaccine formulations interact with the immune system.

In this study, we assessed the immunogenicity of a non-replicative (E1-deleted) CAV2 expressing the rabies virus glycoprotein (RVG) (Cav-G R^0^) with and without the ISA 201 adjuvant in C57BL/6 mice. To monitor the vaccine activity, we quantified the anti-RVG IgG titers by an ELISA and detected the presence of anti-CAV2 antibodies using a Luminex^®^ bead-based immunoassay. Our results showed that ISA 201 improved the immunogenicity against the transgene encoded by Cav-G R^0^ without impacting anti-vector antibody responses in C57BL/6 mice. In addition, we performed an RT-qPCR array to profile the expression of 84 genes related to the antiviral response pathway in the whole blood of mice 24 h after vaccine administration. This approach allowed us to investigate how different CAV2-based vaccine formulations modulate the innate antiviral immune system. The analysis of antiviral gene expression revealed a marked increase in signaling pathways associated with RIG-I-like receptors (RLRs), toll-like receptors (TLRs), NOD-like receptors (NLRs), and interferons (IFNs) in the blood of vaccinated mice.

Overall, our findings demonstrate the potential of ISA 201 as a new adjuvant to improve the efficacy of the CAV2 vector and contribute to the development of more potent vaccine formulations. Moreover, this study deepens our understanding of the innate immune responses elicited by ISA 201 and Cav-G R^0^, paving the way for further advancements in vaccine research and design.

## 2. Materials and Methods

### 2.1. Ethics Statement

All animal experiments conducted in this study were carried out in compliance with ethical standards and regulations, with approval and oversight provided by the ANSES/EnvA/UPEC Ethics Committee (CE2A-16). In addition, these experiments have been authorized by the French Ministry of Research under reference number 12–105, in accordance with French and European guidelines.

### 2.2. CAV2 Recombinant Vaccines and Cells

The construction and production of Cav-G R^0^ vectors have been previously described [14]. These non-replicative E1-deleted vectors expressing RVG were amplified in the DK-E1 cell line. Subsequently, the canine adenovirus vectors were purified by double banding on the CsCl gradient and titrated by end-point dilution. The infectious titers were quantified as a median tissue culture infectious dose per mL (TCID_50_/mL).

The DK-E1 cell line, expressing the CAV2 E1 region, was cultivated as monolayers in Dulbecco’s modified Eagle’s medium (DMEM) supplemented with 4 mM of L-alanyl-L-glutamine dipeptide (GlutaMax^TM^), 1 mM of pyruvate, 10% heat-inactivated fetal calf serum (FCS), 100 IU/mL of penicillin, and 100 μg/mL of streptomycin. The cells were maintained at 37 °C in 5% CO_2_.

### 2.3. Immunization of Mice

Wild-type C57BL/6 mice were obtained from Charles River Laboratories. Fifteen-week-old female C57BL/6 mice were randomly divided into four groups of five mice each. All mice received a single intramuscular dose of either the vaccine or control formulation in the right biceps femoris, with a total volume of 50 µL. Two groups of mice were immunized with 5 × 10^6^ TCID_50_ of Cav-G R^0^ vectors or 5 × 10^6^ TCID_50_ of Cav-G R^0^ vectors mixed with the adjuvant Montanide^TM^ ISA 201 VG (SEPPIC, France). The two control groups received either a physiological saline solution (mock) or the physiological saline solution mixed with ISA 201. ISA 201-containing formulations were emulsified according to the manufacturer’s instructions (50:50 *w*/*w*; temperature, 31 ± 1 °C in both phases; low shear agitation).

### 2.4. Anti-Rabies Antibody Titration

Mouse serum samples were collected at D21 and analyzed for detection and titration of IgG antibodies against the RVG. An indirect ELISA was performed on 96-well microplates coated with RVG (PLATELIA^TM^ RABIES II kit, Bio-Rad, Marnes-la-Coquette, France). Mice sera were pre-diluted in a sample diluent for quantitative analysis. All samples were tested in duplicate, and serum titers were expressed as equivalent units per mL (EU/mL), which is a unit equivalent to the international units defined by seroneutralization [15]. Samples with a titer greater than the seroconversion threshold of 0.5 EU/mL were considered protected against the rabies virus, while samples with a titer below this threshold were considered unprotected. An undetectable seroconversion was defined as a titer below 0.125 EU/mL.

### 2.5. Anti-CAV2 Antibody Responses

Serum samples were collected at D21 and analyzed for the presence of IgG antibodies against CAV2 using Luminex^®^ technology. The viral particles of the CAV2 Manhattan strain were heat-inactivated at 56 °C for 30 min, and a total of 2 × 10^10^ TCID_50_ virions were coupled to 1.25 × 10^6^ carboxylated beads using the Plex^®^ Amine Coupling kit (Bio-Rad, France). The dyed microspheres coated with inactivated CAV2 virus were incubated with 0.5 µL of the mouse sample in blocking buffer containing PBS (pH 7.2), 0.05% TWEEN^TM^ 20 and 5% nonfat dry milk. After the addition of a biotin-conjugated rabbit anti-mouse IgG (ab97044, Abcam, Paris, France) and a streptavidin R-phycoerythrin conjugate (SAPE; 1 μg/mL; Qiagen, Courtaboeuf, France), the quantity of the antibody captured by the antigen was determined by the fluorescence of PE reporter dye using a Luminex^®^ 200 system (Bio-Rad). Fifty beads were counted per region to determine the fluorescent signal. The results were reported as the average fluorescent signal recorded, measured in fluorescence units (FU).

### 2.6. Antiviral Gene Expression Profile

The total RNA was extracted from 250 µL of peripheral whole blood collected 24 h post-immunization, using a Quick-RNA^TM^ Whole Blood kit (Ozyme, France). To eliminate any residual genomic DNA, 5 µg of each RNA sample underwent treatment with the RQ1 DNAse (Promega, Charbonnières-les-Bains, France) and were subsequently purified with an RNeasy^®^ MinElute^®^ Cleanup kit (Qiagen, France). The expression of the antiviral gene was analyzed using an RT^2^ Profiler PCR Array Mouse Antiviral Response (PAMM-122ZF-12, Qiagen, Courtaboeuf, France). For each group of mice, 1 µg of pooled RNA was subjected to reverse transcription using an RT^2^ first-strand kit (Qiagen, Courtaboeuf, France). The PCR set-up included 25 µL of cDNA and RT^2^ SYBR Green Mastermix (Qiagen, Courtaboeuf, France). Real-time PCR was performed on a LightCycler^®^ 480 (Roche Diagnostics, Meylan, France).

The cycle threshold (Ct) values were determined using the “Abs/Quant/2nd Derivative Max” function of LightCycler^®^ 480 software (version LCS480 1.5.1.62, Roche) and analyzed using the Qiagen Data Analysis Center (http://www.qiagen.com/geneglobe, accessed on 2 September 2022).

All samples successfully passed the internal controls of the PCR array, which monitored genomic DNA contamination, first-strand synthesis, and PCR efficiency. The control group consisted of mice that received a mock vaccination. To normalize the Ct values, a set of housekeeping genes (HKGs), including *Actb*, *Gapdh*, *Hsp90ab1*, and *Gusb*, was automatically selected. The selection was based on choosing the set of HKGs in the 96-well plate with the most stable expression levels.

Differential expression analysis of antiviral genes between the vaccinated and control mice was performed using the ∆∆C_T_ method. The fold change (FC) and fold regulation (FR) were calculated based on the ∆∆C_T_ values. For genes showing an up-regulation (FC > 1), the FR was equal to the FC. Conversely, for genes showing a down-regulation (FC < 1), the FR was calculated as the negative inverse of the FC. To determine differentially expressed genes, a threshold of an FC ≥ |2.0| was set. Additionally, a Ct cut-off value of 35 was applied during the analysis.

Differentially expressed genes were visualized using Venn diagrams and heatmaps with genes sorted based on their FC from the Cav-G R^0^-treated group. To establish the connection between genes and their corresponding signaling pathways (as described by Qiagen), a Sankey diagram was constructed utilizing SankeyMATIC (https://sankeymatic.com/, accessed on 8 September 2022). Significantly differentially expressed genes were categorized based on their molecular function, according to the study by McCracken et al. [16], and compared with cell-type-specific gene lists established from human blood by Nakaya et al. [17].

### 2.7. Statistical Analyses

The differences between the two vaccinated groups, Cav-G R^0^ and Cav-G R^0^ with ISA 201, were analyzed using unpaired Mann–Whitney tests. Statistical significance was determined by considering *p*-values ≤ 0.05).

## 3. Results

### 3.1. Vaccination of Mice

Groups of C57BL/6 mice (*n* = 5) were immunized intramuscularly once with 5 × 10^6^ TCID_50_ of Cav-G R^0^ either formulated in a physiologic serum or with ISA 201 (Figure 1a). Two control groups were also included, which received physiological saline solutions with and without ISA 201, respectively. No local reactions were observed upon the palpation after the injections. All mice remained healthy throughout the entire experiment and showed no ill effects or clinical signs.

### 3.2. Antibody Response to Rabies Recombinant Glycoprotein

To assess the immunogenicity of a single intramuscular immunization with 5 × 10^6^ TCID_50_ of Cav-G R^0^ in C57BL/6 mice, individual anti-RVG immunoglobulin G (IgG) concentrations in the sera at D21 were measured using an ELISA (Figure 1b). The control groups, which received physiological saline solutions with and without ISA 201, showed no serological response with anti-rabies IgG titers below the detectable seroconversion threshold of 0.125 EU/mL. In contrast, all of the vaccinated mice had anti-rabies IgG titers above the protection threshold of 0.5 EU/mL, indicating their protection against the rabies virus after a single administration of Cav-G R^0^. The group receiving Cav-G R^0^ and the ISA 201 adjuvant showed a noteworthy increase in their humoral response against the rabies antigen at D21 compared to the group that received Cav-G R^0^ alone. The mean concentration of the anti-RVG IgG in the group receiving Cav-G R^0^ and ISA 201 was 2176 EU/mL, which was significantly higher than the mean of 759.5 EU/mL in the Cav-G R^0^-immunized group (*p* = 0.0317).

### 3.3. Serological Assay for Anti-CAV2 Antibodies

A Luminex^®^-based immunoassay was performed on individual sera at D21 (Figure 2) to assess the impact of ISA 201 on the anti-vector humoral response. Whole CAV2-coated beads were utilized to measure IgG responses directed against all of the CAV2 proteins. Negative controls were implemented on the sera from mock- and ISA 201-treated mice, indicating no detection of the anti-CAV2 IgG. The results demonstrated that all of the Cav-G R^0^-vaccinated mice generated significant levels of anti-CAV2 IgG, with a mean fluorescent signal above 10,000 FU. Furthermore, the addition of ISA 201 did not produce a significant impact on the anti-CAV2 IgG titers (*p* > 0.99), with a mean of 13,368 FU in the Cav-G R^0^-immunized group compared to a mean of 12,645 FU in the Cav-G R^0^- and ISA 201-treated group. These findings suggest that ISA 201 enhances the immunogenicity of the transgene while having no effect on anti-vector antibody responses in C57BL/6 mice.

### 3.4. Analysis of Early Antiviral Immune Responses

In order to evaluate the early systemic innate immune responses triggered by the two vaccine formulations based on Cav-G R^0^, a set of 84 antiviral transcripts was profiled on a pool of whole blood for each group of mice 24 h after vaccination. For the relative quantification, mock-treated mice were used as the control, and differentially expressed genes (DEGs) associated with a Ct value > 35 or −2 < FR < 2 were excluded. A total of 34 significant DEGs were detected across the three groups (Figure 3). Among the Cav-G R^0^-receiving mice, 25 DEGs (14 down-regulated and 11 up-regulated) were identified. These modulations are present as a consequence of successful vaccination and reveal a set of genes involved in the robust systemic antiviral innate immune response to Cav-G R^0^. Of the twenty-seven DEGs (seventeen down-regulated and ten up-regulated) associated with the Cav-G R^0^- and ISA 201- treated group, nineteen overlapped with the Cav-G R^0^-receiving mice (ten down-regulated and nine up-regulated). Furthermore, all modulation signs were conserved, indicating that most of the antiviral transcript modulations induced by the Cav-G R^0^ remained unchanged in the presence of the ISA 201 adjuvant. The ISA 201 adjuvant was found to abrogate the modulation of six genes induced by the Cav-G R^0^ vector while reinforcing the modulation of eight other genes (seven down-regulated and one up-regulated). Only five DEGs (three down-regulated and two up-regulated) were related to the ISA 201 formulated in the saline solution. Among them, four were found to overlap with the Cav-G R^0^ + ISA 201-treated group. Overall, these results reveal a marginal effect of the adjuvant on antiviral innate immune responses in blood 24 h post-vaccination of mice.

### 3.5. Pathways and Cell Types Prediction Related to Differentially Expressed Genes

To further elucidate the impact of transcriptional modifications of the whole blood cells on biological functions, the DEGs were examined according to the function or signaling pathways of their products (Figure 4). Genes coding for innate immune activators, inhibitors, and adaptors, such as TRAF3; cytokines, such as CCL5; enzymes; MAP kinases; and transcription factors, including IRF3, RELA, and NFKB1, were down-regulated by the Cav-G R^0^ vector. On the other hand, genes coding for cytokines, such as CXCL10; receptors, such as DDX58, DHX58, and TLR7; transcription factors, such as STAT1 and IRF7; enzymes, such as MX1; and other immune regulators, including ISG15 and MEFV, were up-regulated by the Cav-G R^0^ vector (Figure 4a). Down-regulated genes were mainly involved in signaling downstream of the RLRs (ten genes) and the TLRs (six genes), while up-regulated genes were more related to type I IFN (IFN-I) signaling and responsive genes (six genes), the TLRs, RLRs, and chaperones (four genes), and the TLR, RLR, and NLR responsive genes (four genes) (Figure 4b). A comparison with cell-type-specific gene lists established from human blood suggests that up-regulated genes in the CAV-G R^0^-treated mice are preferentially expressed in dendritic cells (DCs) (one gene in the pDCs and two genes in the mDCs) and monocytes (four genes). Down-regulated genes were preferentially expressed in NK cells (one gene), B cells (one gene), T cells (four genes), and monocytes (six genes) [17]. Adjuvantation slightly increased the number of down-regulated genes preferentially expressed in the T, NK, and B cells (Appendix A).

## 4. Discussion

Adenovirus-vectored vaccines have emerged as a highly promising platform for developing new vaccines. The development process involves several steps, including optimizing the antigen, vector structure, route of administration, and vaccine dosage. Adjuvants are often added to improve the immunogenicity of recombinant antigens. While adjuvantation is not commonly used with viral vectors, it can be highly beneficial. For example, the use of poly-ICLC as an adjuvant with Ad5-FMD has been shown to enable an 80-fold vaccine dose-sparing effect [18]. Promising results have also been reported in vaccination trials that combined Ad5 with ISA 206 [9,10,11].

We have previously demonstrated that the Cav-G R^0^ vector elicits a substantial titer of rabies virus-neutralizing antibodies (>10 IU/mL) in C57/Bl6 mice, resulting in a 100% survival rate among vaccinated mice upon a lethal challenge [19]. In this study, we investigated the adjuvant effect of ISA 201 in combination with Cav-G R^0^ in mice for the first time. Our results showed that vaccination with Cav-G R^0^ induced high anti-rabies antibody titers that were associated with protective immunity three weeks later. Moreover, the water-in-oil-in-water formulation of ISA 201 enhanced specific IgG titers against the RVG transgene encoded by Cav-G R^0^, with a good safety profile. While previous studies have confirmed the efficacy and safety of ISA 201 in potentiating immune responses with recombinant antigens [12,13,20,21], our study provides evidence of its potential benefits for live recombinant viral-vectored vaccines. In contrast, Zajac et al. reported a marginal increase in the IgG response to the pp62 antigen when an experimental African swine fever virus vaccine, based on the human adenovirus, was combined with ISA 201 [22]. Nonetheless, the interpretation of this finding is hindered by several factors, including the use of a cocktail of 42 adenoviral vectors expressing multiple ASFV antigens and the potential for interference with the host’s immune system. Although the necessity of ISA 201 in our study is uncertain, it could be a crucial advantage for fewer immunogenic antigens and lower doses of the recombinant vector [11] or to extend the duration of protective immunity in other animal species, such as cattle, swine, and small ruminants. Therefore, the combination of a CAV2-based vector and ISA 201 may prove to be a valuable tool for developing vaccines against major diseases that threaten production animals, such as foot-and-mouth disease (CAV2-ΔE3-VP1 [23] and CAV2-ΔE1-P1/3C [8]), porcine reproductive and respiratory syndrome (CAV2-ΔE3-GP5 [4]), peste des petits ruminants (CAV2-ΔE3-H) [5]), or bluetongue (CAV2-ΔE1-VP7 [3]).

In our study, we found an interesting disconnection between the magnitude of immune responses directed toward the transgene and the vector itself. This discrepancy may be related to the nature of the antigens, their presentation to the immune system (whether they are injected or produced by the transduced host cells), and their biodistribution, as well as their amount. CAV2-vectored vaccines retain their viral properties to enter target cells, engage intracellular trafficking to deliver their genome, and promote antigen production. Moreover, the use of potent promoters, like CMV [19], enhances transgene transcription in vivo, which leads to sustained antigen expression compared to the recombinant protein that is quickly degraded and limited to the inoculated dose. A prolonged expression of the vaccine antigen has been shown to maintain activated CD8^+^ T cells [24,25]. Double emulsions, like the Montanide^TM^ ISA 201 VG used in our study, are known to sustain the release of antigens. Therefore, it is possible that the adjuvant increases the duration of the expression of the targeted antigen by modulating its delivery. However, to evaluate the mode of antigen presentation and the early mechanism of action of this combination, it is necessary to explore the site of injection and the draining lymph nodes, as we can only observe the downstream consequences of the stimuli in the blood.

Innate immunity plays a crucial role in initiating desirable transgene-specific adaptive immune responses [26]. In order to gain a better understanding of the specific contributions of the CAV2 vector and ISA 201 in this process, we conducted a comprehensive analysis of the systemic innate antiviral transcript modulations induced by CAV2-based vaccine formulations. For the first time, we examined the expression of antiviral genes in whole blood samples collected from mice 24 h after vaccination.

Our analysis revealed a distinct Cav-G R^0^ gene expression pattern, characterized by the up-regulation of nine specific genes, including *Ddx58, Il15*, *Mefv*, *Dhx58*, *Stat1*, *Mx1*, *Cxcl10*, *Isg15*, and *Irf7*. Interestingly, these genes were also found to be up-regulated in human subjects vaccinated with Merck Ad5/HIV [27], as well as in sheep vaccinated with Ad5-FMDV, with the exception of *STAT1* [9]. Similarly, in macaques, the genes *STAT1*, *MX1*, *CXCL1*, and *IRF7* showed up-regulation in blood samples taken 1 day after vaccination with ChAd155-RG [28] and 16 h after the transduction of ovine DC subsets with the CAV2 vector [29]. These findings highlight the occurrence of these blood gene signatures across various mammalian hosts, indicating that they may be attributed to the conserved properties of adenoviral vectors.

While the effect of ISA 201 on the antiviral immune response targeted by our study was modest, the few modulations observed suggest an effect of ISA 201 on the blood transcriptome of mice. To further investigate the role of ISA 201, broader studies using whole transcriptome sequencing could be conducted to identify non-antiviral gene signatures and other relevant biological functions. It can also be hypothesized that the main effect of the adjuvant is to trigger the recruitment of immune cells at the injection site and the draining lymph nodes, which warrants further investigation.

The early up-regulation of genes by Cav-G R^0^ in the whole blood of mice is linked to cellular innate signaling pathways involving RLRs, TLRs, NLRs, and IFN-I (Figure 4b). While the molecular biology of CAV2 is still poorly described, these pathways are prominently activated during adenoviral entry, underscoring their significance [30], whereas the down-regulated genes are more closely related to RLR and TLR downstream signaling, which could reveal the presence of negative feedback mechanisms. Our findings suggest that Cav-G-R^0^ can be detected by pattern recognition receptors within hours of injection, leading to the stimulation of IFN-I and ISGs. Notably, we observed a significant up-regulation of *Irf7*, while *Irf3* was down-regulated 24 h after vaccination. Both *Irf7* and *Irf3* are regulated by the initial production of IFN-I: IRF7, acting as a key regulator of IFN gene expression, while IRF3 probably cooperates with IRF7 for optimal activity. Since IRF7 is responsible for a positive feedback loop that amplifies the initial IFN-I response [31], its up-regulation may last longer.

Although IFN-Is play a critical role in antiviral innate immunity, their specific contribution to the establishment of a protective immune response is still unclear. It has been observed that the magnitude and duration of antigen expression are favorable for vaccine efficacy [1]. However, excessive production of IFN-I can have deleterious effects by impairing transgene expression [32,33]. Furthermore, it has been demonstrated that early inhibition of IFN-I leads to the improved long-term efficacy of viral vaccines [34]. Paradoxically, IFN-I has also been shown to enhance the efficacy of influenza vaccines [35] and adenoviral vector vaccines [36,37]. Therefore, early induction of interferon responses may have a detrimental effect on transgene expression, while later expression of the vector-encoded IFN-I could enhance immune responses against the vaccine antigen. Consequently, it is crucial to distinguish between immune responses directed against the vector (from the early hours post-inoculation) and those targeting the vaccine antigen (following the expression of viral genes).

In this study, we also highlight the up-regulation of genes that are stimulated by interferons. Among their related proteins, ISG15 plays a crucial role during infection across a wide range of viruses. It functions by antagonizing viral replication, modulating IFN-I signaling, and regulating cytokine release. ISG15 also influences the function of various immune cells due to its immunomodulatory and cytokine properties. For instance, it induces the maturation of DCs, recruitment and activation of neutrophils, stimulation of macrophages, and proliferation of NK cells, and promotes interferon-γ production [38]. Additionally, CXCL10, also known as interferon γ-induced protein 10 (IP-10), is a chemokine secreted by cells stimulated with IFNs. It acts as a chemoattractant for monocytes and activated T cells [39].

Since blood serves as a transit compartment for immune cells, the observed gene modulations could be attributed to changes in the proportion of immune cells actively migrating following intramuscular injection. By comparing with cell-type-specific gene lists, we identified up-regulated genes that were predominantly expressed in DCs and monocytes, while down-regulated genes were more commonly expressed in NK cells, B cells, T cells, and monocytes [40]. This pattern of gene expression aligns with previous findings observed in other adenoviral vaccine vectors, suggesting that it may be a consequence of the early migration of immune cells to the injection site or draining lymph nodes [27,28]. Further investigations are necessary to determine whether the adjuvant employed in this study could enhance the relocation of cellular effectors.

## 5. Conclusions

This study presents novel findings regarding the Montanide^TM^ ISA 201 VG adjuvant to enhance the immune response against a vectored antigen while maintaining balanced immunogenicity against the vector itself. By demonstrating the potential of this new adjuvant in improving the efficacy of the CAV2 vector in mice, these results contribute to the development of more potent vaccine formulations for relevant animal species like swine and sheep.

Moreover, the characterization of early transcriptomic antiviral responses in blood sheds light on the underlying mechanisms driving the efficacy of CAV2-vectored vaccine. Additionally, it provides valuable insights into the benefits of ISA 201. Furthermore, by comparing these findings with other studies, the characterization of blood transcriptomic signatures holds the potential to identify predictive biomarkers for vaccine efficacy. Collectively, these data facilitate the rational design of adenovirus-based vaccines, thereby advancing the progress of vaccine platforms.

## Figures and Tables

**Figure 1 viruses-15-01664-f001:**
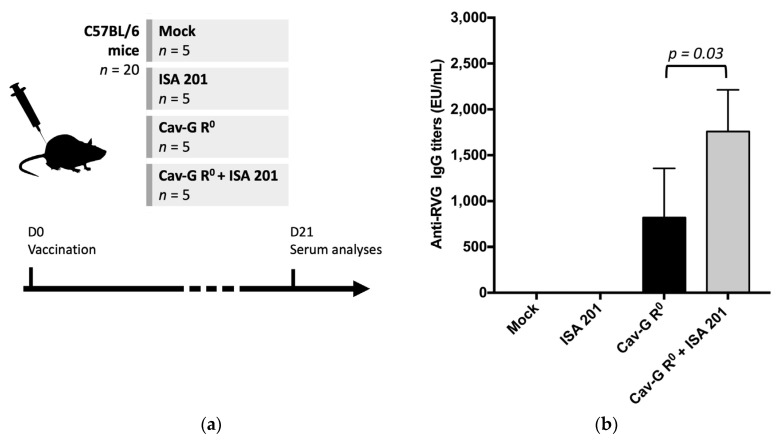
Serum anti-RVG antibody levels following vaccination of mice with Cav-G R^0^ and/or ISA 201. (**a**). Experimental design. Four groups of 5 mice each were immunized intramuscularly with 5 × 10^6^ TCID_50_ of Cav-G R^0^ either with or without ISA 201. Sera were collected at D0 and D21 post-vaccination. (**b**) Antibody responses against the RVG protein were evaluated using an ELISA (Platelia^TM^ Rabies II Assay). Statistically significant difference between the two groups was calculated using unpaired Mann–Whitney test.

**Figure 2 viruses-15-01664-f002:**
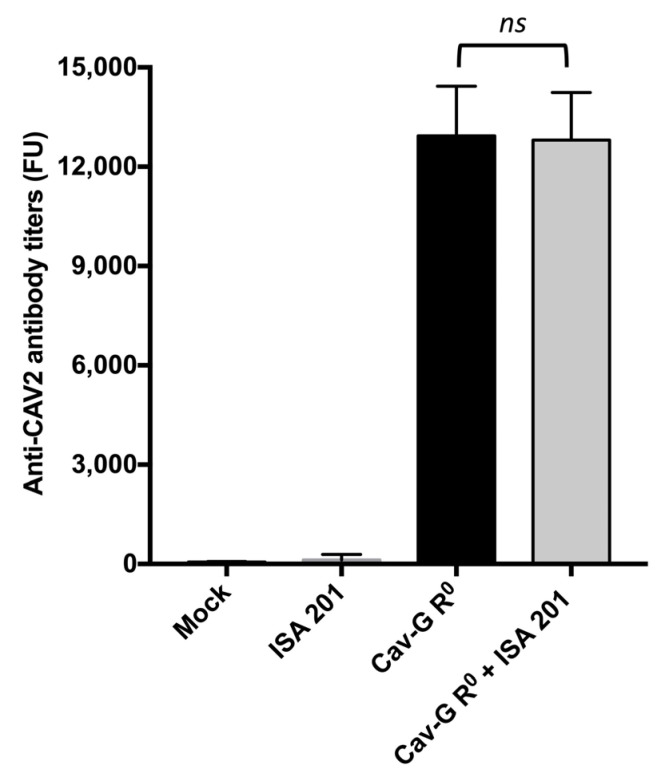
Serum anti-CAV2 vector antibody levels following vaccination of mice with Cav-G R^0^ and/or ISA 201. The antibody titers against the CAV2 vector were measured using a Luminex^®^ bead-based immunoassay. Statistical differences between the two groups were determined using the unpaired Mann–Whitney test (ns: not significant).

**Figure 3 viruses-15-01664-f003:**
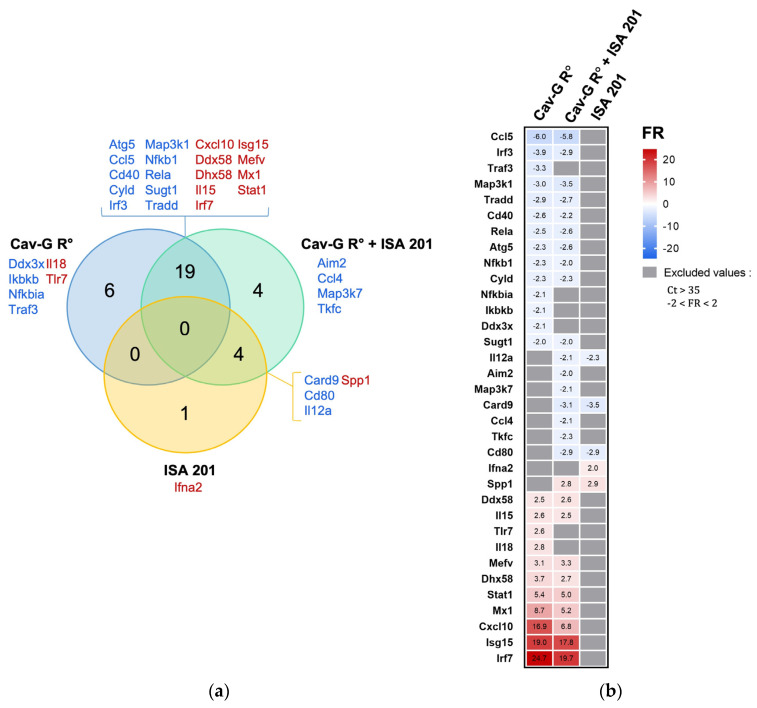
Modulation of antiviral genes in blood 24 h post-vaccination with Cav-G R^0^ and/or ISA 201. The fold regulation (FR) of gene expression was calculated by comparing one pool of whole blood from each treated mouse group with a reference of saline solution. (**a**). The Venn diagram illustrates the comparison of DEGs. The overlapping regions between the two circles represent the common DEGs shared between the two treatments. (**b**). A heatmap displays significant DEGs. Up-regulated genes are represented by red colors, while down-regulated genes are indicated by blue colors, compared to the mock group. Grey denotes excluded values with a Ct > 35 or −2 < FR <2.

**Figure 4 viruses-15-01664-f004:**
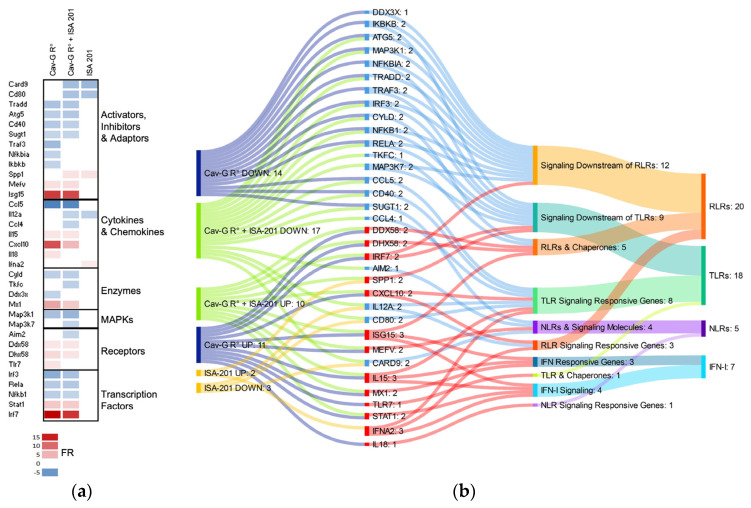
Functional enrichment analysis of antiviral gene modulations in blood 24 h after vaccination with Cav-G R^0^ and/or ISA 201. The fold regulation (FR) of gene expression was calculated by comparing one pool of whole blood from each treated mouse group with a reference of saline solution. Significant values are represented by red colors, indicating up-regulated gene expression, and blue colors, indicating down-regulated gene expression, in comparison to the mock group. (**a**) The heatmap displays the clustering of genes based on their molecular function. This classification follows the proposal by McCracken et al. [16]. (**b**) The Sankey diagram reveals functional groups of significant DEGs that are involved in various antiviral response pathways. This classification is based on the description provided by Qiagen as part of the RT^2^ PCR array.

## Data Availability

Data sharing is not applicable to this article.

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
