# Peer review of "An Oil-Based Adjuvant Improves Immune Responses Induced by Canine Adenovirus-Vectored Vaccine in Mice"

_viruses, 2023, doi:10.3390/v15081664_

Round 1

Reviewer 1 Report

The manuscript reported that ISA 201 improves immune responses induced Cav-G R0.

Although the current research focus on canine adenovirus vectors has declined, it is still a good choice to find suitable adjuvants to promote vaccine research.The writing idea is clear and organized.I clearly recommend the manuscript for publication in Viruses.

Minor Comments:

1.     Whether ISA 201 only plays a role in promoting immunity to CAV2, or whether it has an effect on other vectors.

2.     Whether anti-RVG immunoglobulin G (IgG) to resist virus infection. Lack of your virus challenge test validation

3.     Adjuvantation slightly increased the number of type I IFN (IFN-I) signaling and responsive genes. It would be better if we could test for the relevant factors by ELISA.

Author Response

Dear reviewer,

Thank you for your valuable feedback on our manuscript. We appreciate your insightful comments and suggestions (which have helped us enhance the quality of our work). In this response, we address each of your comments and provide the necessary clarifications.

1) Whether ISA 201 only plays a role in promoting immunity to CAV2, or whether it has an effect on other vectors.

The ISA 201 VG adjuvant finds primary application in formulations with inert antigens. Nevertheless, it has also been tested with inactivated recombinant virus (An inactivated recombinant rabies virus displaying the Zika virus prM-E induces protective immunity against both pathogens - 10.1371/journal.pntd.0009484) or bacteria (A bacterium-like particle vaccine displaying Zika virus prM-E induces systemic immune responses in mice - 10.1111/tbed.14594), and VLPs (Large-scale manufacture of VP2 VLP vaccine against porcine parvovirus in Escherichia coli with high-density fermentation - 10.1007/s00253-020-10483-5) as antigens. To our knowledge, the only published study in which ISA 201 has been used with a live recombinant viral vector involves a human adenovirus type 5 (Immunization of pigs with replication-incompetent adenovirus-vectored African swine fever virus multi-antigens induced humoral immune responses but no protection following contact challenge - 10.3389/fvets.2023.1208275).

In this study, the efficacy of a vaccine comprising a cocktail of 42 recombinant adenoviruses expressing ASFV antigens was evaluated. The results revealed a marginal increase in the humoral response to one of the viral antigens in the presence of ISA 201. However, the interpretation of this observation is hindered by several factors, including the use of numerous viral vectors expressing multiple ASFV antigens and the potential for interference with the host immune system.

2) Whether anti-RVG immunoglobulin G (IgG) to resist virus infection. Lack of your virus challenge test validation

We conducted a previous assessment of the vaccine efficacy using the Cav-G R0 vector in mice, followed by a challenging phase. The results obtained from this study revealed that the expression of RVG resulted in complete protection, with 100% of vaccinated animals surviving after exposure to a virulent strain of the rabies virus through intracranial administration. (Canine Adenovirus Based Rabies Vaccines - PMID: 18634509).

3) Adjuvantation slightly increased the number of type I IFN (IFN-I) signaling and responsive genes. It would be better if we could test for the relevant factors by ELISA.

We appreciate your suggestion regarding testing for the relevant factors by ELISA to further investigate the effects of adjuvantation on type I IFN (IFN-I) signaling and responsive genes. However, at this stage of our research, we did not incorporate ELISA assays to directly measure the levels of specific IFN-I signaling molecules. Our focus in this study was on examining the changes in gene expression profiles rather than quantifying the actual protein levels. While we acknowledge that ELISA would provide more quantitative data on IFN-I signaling factors, our research design and available resources limited us to gene expression analysis. We believe that studying the changes in gene expression patterns provides valuable insights into the potential regulatory mechanisms associated with adjuvantation. We appreciate your suggestion and acknowledge the potential benefits of incorporating ELISA in future studies to complement our findings. We will consider this suggestion for future research projects in order to provide a more comprehensive understanding of the effects of adjuvantation on IFN-I signaling.

Reviewer 2 Report

The paper titled "An oil-based adjuvant improves immune responses induced by canine adenovirus vectored vaccine in mice" focuses on evaluating the immunogenicity of a non-replicating CAV2 vector encoding the rabies virus glycoprotein (RVG) with and without the oil-based adjuvant Montanide ISA 201 VG. The authors demonstrate that the inclusion of the adjuvant enhances the humoral immune response against the vaccine antigen RVG. Additionally, the study provides insights into the early antiviral responses induced by the vaccine vector at the transcriptomic level. This research addresses a significant research gap by investigating the impact of an oil-based adjuvant on the immunogenicity of a CAV2-based vaccine vector. The comprehensive analysis of innate immune responses and gene expression offers valuable insights into the underlying mechanisms of the vaccine's efficacy. The characterization of blood transcriptomic signatures also holds potential for identifying predictive biomarkers for vaccine efficacy. The paper is well-structured, providing clear descriptions of the methodology, results, and interpretation of the findings.

I have a few minor questions regarding this manuscript. Here they are:

1) Did the study measure the presence of neutralizing antibodies against RVG?

2) It would be more convincing if there were results from RAV challenge experiments to assess protective efficacy.

3)The study primarily focuses on humoral responses and the early antiviral responses induced by the vaccine vector at the transcriptomic level, with limited assessment of cellular immune responses or functional assays.

Author Response

Dear reviewer

Thank you for your valuable feedback on our manuscript. We appreciate your insightful comments and suggestions (which have helped us enhance the quality of our work). In this response, we address each of your comments and provide the necessary clarifications.

1) Did the study measure the presence of neutralizing antibodies against RVG?

The PLATELIATM RABIES II kit is an ELISA (Enzyme-Linked Immunosorbent Assay) based on a highly purified glycoprotein G, which plays a crucial role in the production of neutralizing antibodies. The detection of neutralizing rabies antibodies in blood is considered a reliable indicator of successful vaccination and adequate protection against rabies. This ELISA has been specifically developed to measure the levels of rabies anti-glycoprotein G antibodies in post-vaccination serum samples. It is a validated and widely recognized ELISA method. During the validation process, a strong correlation was established between the PLATELIATM RABIES II test and other neutralization assays such as the RFFIT and the FAVN. This correlation was observed not only in animal sera samples but also in human sera samples, further confirming its reliability and accuracy. Furthermore, it should also be mentioned that the PLATELIATM RABIES II assay has been certified by the World Organization for Animal Health (WOAH), underlying its recognition and acceptance within the scientific and veterinary community.

2) It would be more convincing if there were results from RAV challenge experiments to assess protective efficacy.

We conducted a previous assessment of the vaccine efficacy using the Cav-G R0 vector in mice, followed by a challenging phase. The results obtained from this study revealed that the expression of RVG resulted in complete protection, with 100% of vaccinated animals surviving after exposure to a virulent strain of the rabies virus through intracranial administration. (Canine Adenovirus Based Rabies Vaccines - PMID: 18634509).

3)The study primarily focuses on humoral responses and the early antiviral responses induced by the vaccine vector at the transcriptomic level, with limited assessment of cellular immune responses or functional assays.

You raise a crucial point, and we completely agree that investigating cellular immune responses and conducting functional assays would enhance the comprehensive understanding of our vaccine vector's efficacy. We would like to point out that we have already analyzed the cell-mediated immune responses induced by the Cav-G R0 vaccine. In a separate and preliminary experiment, we conducted a lymphocyte proliferation assay on the lymph nodes of immunized sheep, and the results showed significant RVG-specific CD8+ and CD4+ T cell responses. This study, titled "Canine adenoviruses elicit both humoral and cell-mediated immune responses against rabies following immunization of sheep" (doi:10.1016/j.vaccine.2010.11.068), provided valuable insights into the cellular immune responses elicited by the vaccine.

However, in the present report, our primary objective was to examine the transcriptomic changes and molecular mechanisms associated with the humoral responses and early antiviral responses induced by the vaccine vector. Our focus was to gain a deeper understanding of the gene expression patterns and regulatory pathways involved in the immune response following vaccination.

In response to your suggestion, we are currently planning to extend our research to include assessments of cellular immune responses, such as measuring T cell activation, cytokine production, and cytotoxicity. Additionally, we intend to perform functional assays to evaluate the efficacy of the cellular immune response in combating viral infections. By incorporating these aspects into our study, we aim to provide a more comprehensive evaluation of the vaccine vector's immune responses, which will significantly contribute to a better understanding of its overall effectiveness.